# Sustainability in Universities: DEA-GreenMetric

**Rosa Puertas and Luisa Marti *** 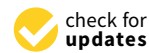

Department of Economics and Social Sciences, Universitat Politècnica de València, Valencia 46022, Spain
* Correspondence: mlmarti@esp.upv.es; Tel.: +0034-963877000

**Abstract:** Many universities are currently doing important work not only on environmental issues, but also on social and economic matters, thereby covering the three dimensions of sustainability. This paper used data envelopment analysis to construct a synthetic indicator based on the variables that make up the UI GreenMetric. The aim was to quantify the contribution of universities to sustainability, rank all campuses accordingly, and evaluate specific aspects of their related institutional policies. First, cluster analysis was applied, yielding four homogeneous groups of universities. DEA was then applied to these clusters in order to construct the synthetic indicator. The proposed indicator, DEA-GreenMetric, revealed that the USA and the UK were the countries that were home to the greatest number of universities actively involved in all aspects of sustainability. In addition, this new index provides a complete ranking of universities, circumventing the issue of the duplicate scores assigned by UI GreenMetric. Finally, it can be seen that greater efforts are required for universities to improve their performance relating to environmental variables (energy, water use, and waste treatment) than to make improvements in infrastructure, transport, or education.

**Keywords:** universities; sustainability; GreenMetric

## 1. Introduction

As entities dedicated to the transfer of knowledge and promotion of research on a wide range of subjects, universities play an important role in the socio-economic development countries. Moreover, they give students the skills and abilities needed to integrate into the labor market. This has prompted the emergence of a variety of university rankings (QS World Ranking, Shanghai Ranking, The World University Ranking, SCImago Institutions Ranking, among others). The objective of all of these ranking schemes is to evaluate the universities' academic and research reputation or their performance, with issues related to environmental protection and sustainability assigned far less importance.

The concept of sustainability dates back to 1987; at that time, its scope was limited to the relationship between people's aspirations for a better life and the constraints on this imposed by nature [1]. The idea has since been expanded and adapted in response to the changes witnessed in the intervening years. The study in Bell and Morse [2] explained that in its original form, sustainability was primarily associated with the maintenance of environmental quality; other elements were subsequently incorporated to give rise to the current concept of sustainability that is comprised of three dimensions: social, economic, and environmental [3–5]. As Castellani and Sala [5] state, sustainability is not a universally-accepted concept, nor is there a single piece of legislation that enables its equal implementation in all countries and social spheres. Rather, it can be shaped by individual contexts, with different weights thus assigned to the three dimensions.

At both the national and international levels, strategies for implementing sustainability are being redefined on the basis of international programs and networks such as the United Nations Sustainable Development Solutions Network, the International Sustainable Campus Network, the Association for the Advancement of Sustainability in Higher Education in the United States, the Environmental

Association for Universities and Colleges in the United Kingdom, or the University Impact Ranking. Another aspect that is no less important, and yet has received far less attention from researchers, concerns the measurement instruments that enable analyses of the level and scope of sustainability achieved. Some indices have been developed to evaluate specific areas of sustainability such as the Ecological Footprint proposed by Wackernagel and Rees [6], the Living Planet Index [7], the Environmental Sustainability Index [8,9], and the Human Sustainable Development Index [10], among others.

In the field of higher education, different approaches to sustainability have been taken over the years. Even before the current century, sustainability formed part of the universities' institutional vision and practice. The literature contains studies such as Van Weenen [11], which outlines the meaning of sustainable development in order to provide guidelines and advice for shaping university strategies and practices. Furthermore, Sharp [12] concludes that the environmental imperative calls for a rapid and far-reaching response from the university sector, one that goes well beyond what we have seen to date. At the same time, Shriberg [13] claims that cross-institutional tools for assessing sustainability in higher education are rapidly emerging, an idea that has recently been reinforced by the use of indices associated with universities.

A key aspect of sustainability in universities is the transportation of students and workers to and around campus. The fact that these institutions tend to be located on the outskirts of the city creates transportation needs for all of their users. Balsas [14] concluded that in order to create more bicycle and walking-friendly campuses, efforts should be focused on the following seven elements: transportation demand management strategies, organization, planning, facilities, promotion, education, and enforcement.

More recent research has examined sustainability in universities in general terms [15,16], with some studies taking a holistic view [17], or have highlighted the participation of university stakeholders in the development of sustainability [18].

In addition, there is growing recognition of the work done by universities to incorporate sustainable development into their day-to-day activities [19–22]. These organizations have the required capacity and instruments to anticipate change and be proactive in implementing organizational reforms aimed at achieving greater sustainability [23,24]. Thus, various studies have provided definitions of the sustainable university. They agree on three essential elements: the universities' duty to safeguard the environment, ensure social justice, and develop sustainable economic growth [25,26].

The high level of activity on university campuses, together with growing concern about climate change, has created a need to analyze their environmental impact in order to mitigate adverse effects. In this respect, a number of indices have been developed to quantify the contribution made by these institutions; these include the Green League 2007 and the Environmental and Social Responsibility Index 2009 [27]. However, these indices have not had the expected impact; consequently, in 2010, Universitas Indonesia (UI) developed a worldwide ranking of "green" universities, with the aim of evaluating their commitment to all aspects of sustainability. This index, called the UI GreenMetric, has been used as an instrument to support the sustainable development of universitie', as can be seen in the studies of Suwartha and Sari [28] and Sonetti et. al [29]. In particular, the latter study used this index to compare an Italian and a Japanese university. In more recent studies, Drahein et. al [30] applied the UI GreenMetric to analyze sustainability in Brazilian universities, while Parvez, and Agrawal [31] did the same for higher education institutions in India. In addition, Ragazzi and Ghidini [32] proposed possible methodological improvements to the construction of the index.

This study proposes, first of all, an alternative index constructed from the variables used in the UI GreenMetric. It is a composite indicator developed using data envelopment analysis (DEA), a methodology that allows all universities to be ranked according to their contribution to sustainability. Moreover, the use of DEA resolves the issue of the duplicate scores that can be found in the UI GreenMetric; the approach to weighting established by UI means that two different universities can obtain the same Total Score. Second, the aim was to identify potential critical factors for campus

sustainability. This will enable campuses to orient their institutional policies toward the elements that require closer attention.

In order to achieve these objectives, we first carried out a cluster analysis, which allowed us to identify homogeneous groups of universities according to the variables that determine their sustainability performance; this grouping is a prerequisite for the correct application of DEA. Subsequently, we calculated the synthetic indicator for the groups created by using cross-efficiency (CE) to rank the efficient observations. The Kruskal–Wallis statistic allowed us to determine whether the different subsamples created according to a certain value of the synthetic indicator differed significantly from each other in terms of their mean inputs and outputs. The results of this test enabled a better characterization of these subsamples. Finally, the targets calculated by means of DEA can be used to identify potential areas of improvement for the analyzed universities.

This study represents a novel contribution to the literature that offers global rankings of university campus sustainability. As well as providing a ranking, the proposed index helps us understand more about the campuses that contribute most actively, specifying the area of sustainability that is the focus of their greatest efforts.

The rest of the article is structured as follows. Section 2 explains the methodologies used: cluster analysis, DEA and CE. Section 3 details the samples created by means of the cluster analysis and the variables used to construct the synthetic indicator. Section 4 presents the results obtained. Finally, Section 5 summarizes the main conclusions of the article.

## 2. Methodology

The proposed research requires the application of clustering and DEA techniques in order to construct the synthetic index. Both of these techniques have been widely used in the field of higher education, albeit with different objectives. Indeed, the literature contains a number of studies in which cluster analysis has been used to create homogeneous samples in order to obtain robust results in the subsequent application of the DEA methodology [33–36]. In the same vein, this study presents a cluster analysis where the six categories of the UI GreenMetric were used to classify the universities in the sample into homogenous groups in terms of their level of sustainability.

This is a multivariate statistical technique that facilitates the grouping of elements, aiming to achieve not only the maximum within-group homogeneity, but also the greatest inter-group difference. In the first stage, an agglomerative hierarchical clustering algorithm is applied, starting with a situation where each observation constitutes its own cluster. Then, in successive steps, clusters are merged until the appropriate number of clusters has been reached, with the squared Euclidean distance between clusters taken as the agglomerative criterion. The application of this technique allows the researcher to determine the optimal number of groupings there should be in the sample, a number which is a priori unknown. From all of the available hierarchical algorithms, Ward's Method was selected for this article; according to Kuiper and Fisher [37], this is a powerful classification technique that merges different elements while trying to minimize the within-cluster variance. DEA was then applied to the homogeneous groups in order to construct the synthetic index.

The concept of efficiency was introduced in the literature by Farrell [38], who proposed the use of a production function to determine a company's level of efficiency. This idea was later generalized by Charnes and Cooper [39], who developed the DEA methodology to measure the relative efficiency of a set of observations (decision making units, DMUs) defined by multiple inputs and outputs [40]. DEA is a nonparametric technique that determines the maximum efficiency of each of the DMUs relative to the level reached by the other units of the study, so it is important that the sample analyzed is homogeneous. Efficiency is calculated as the ratio between the weighted sum of the products obtained and the resources used to obtain them, by means of the following equation:

$$Max\ E_j = \frac{\sum_{r=1}^{s} u_r y_{rj}}{\sum_{i=1}^{m} v_i x_{ij}} \quad \forall\ j = 1, \ldots n \tag{1}$$

s.a.

$$0 \leq \frac{\sum_r u_r y_{rj}}{\sum_i v_i x_{ij}} \leq 1 \quad \forall j = 1, \ldots, n$$
$$u_r, v_i > 0 \; \forall r = 1, \ldots, s; \; i = 1, \ldots, m$$

where

*m* and *s:* number of input and output variables, respectively

$y_{rj}$: *j*th university output *r* (DMUj)

$x_{ij}$: input *i* from *j*th university (DMUj)

$u_r$: weight assigned to output *r*

$v_i$: weight assigned to input *i*

*n*: number of universities.

The restrictions imposed on the target function limit the value of efficiency (i.e., between 0 and 1) by assigning a value of 1 to fully efficient DMUs and preventing negative weights from being used. Solving this linear programming problem for each of the observations allowed us to calculate the set of weights, u and v, that award the maximum efficiency to each DMU (to do so, the original model must first be linearized and converted to a dual problem). In this model, increases in the volume of inputs yield proportional increases in outputs; it is thus referred to as DEA under constant returns to scale (CCR).

The approach with variable returns to scale leads to the efficient frontier forming a convex zone where the locations of all the points are more limited than those with constant returns to scale, thus yielding equal or greater efficiency results.

In short, DEA allows us to classify the DMUs of a sample as efficiently or inefficiently according to their location with respect to the so-called efficient production frontier. Thus, those universities that have the best sustainability practices in comparison to the others in their group form the said frontier, making it possible to determine which aspects the other universities should modify in order to improve their position.

The DEA methodology makes it possible to differentiate between efficient and inefficient observations, but does not provide a ranking of the efficient observations. Thus, the use of CE is needed to achieve a complete ranking of all of the efficient observations. CE was originally proposed by Sexton et. al [41] and then further developed by Doyle and Green [42] in order to overcome the main limitations of DEA. As highlighted by Angulo-Meza and Lins [43], these limitations include not only the inability to distinguish between efficient units, but also the fact that an inappropriate weighting scheme can distort the results. CE is used to assess the performance of each university, computed using the input and output weights that are optimal for the other institutions. The resulting CE matrix contains information on the efficiency of a campus relative to its peers. This allows the researcher to rank all of the observations that have an efficiency score of 1. Each element is calculated by means of the following expression:

$$E_{kj} = \frac{\sum_{r=1}^{s} u_{rk} y_{rj}}{\sum_{i=1}^{m} v_{ik} x_{ij}} \quad j = 1, \ldots, n; \; k = 1, \ldots, n \tag{2}$$

where $u_{rk}$ and $v_{ik}$ are the optimal multipliers obtained by DEA for the corresponding university, with the original efficiency scores on the diagonal.

Thus, the value of $E_{kj}$ is obtained by evaluating university j using the optimum weights for university *k*. The DeaR software [44] was used to calculate the efficiency levels of each of the universities analyzed.

In this context, the DEA technique analyzes the level of efficiency with which the inputs enable the achievement of certain outputs. However, this method is a powerful tool that can be applied to other types of multidimensional analyses including the construction of synthetic indices [45]. Such indices, which date back to the early 1990s, feature prominently in the literature [46–53]. In the process of

constructing the synthetic indicator, traditional inputs are replaced by variables of a negative nature (the higher the value, the worse the result), while the outputs are represented by positive variables. By maximizing Equation (1) for a given university, we calculated the set of weights that assigned the highest possible value of the synthetic indicator to that observation.

Once the results of the synthetic indicator were obtained, the Kruskal–Wallis test was applied to determine whether the mean inputs and outputs of the most efficient observations were significantly different from those of the least efficient observations. This enables a better characterization of the sustainability of the universities.

## 3. UI GreenMetric Variables and Cluster Analysis

The UI GreenMetric World University Ranking was first published in 2010 by UI; the goal was to be able to assess the level of sustainability of higher education institutions. At that time, a total of 95 universities from around the world took part, and by 2018 this figure had risen to 719 campuses. The ranking takes into account the three dimensions of sustainability: environment, economy and equity. The environmental dimension includes the use of natural resources, environmental management, and pollution prevention; the economic dimension focuses on cost savings and benefits; while the social dimension centers on education, community, and social participation. As stated in Guideline of UI GreenMetric World University Ranking [54], in addition to measuring universities' efforts to improve the sustainability of their different campuses, the basic objectives of this ranking are as follows:

- To stimulate academic debate on sustainability in education and the greening of university institutions.
- To make universities the standard-bearers for sustainability goals and disseminate these to society.
- To provide a comparative tool for assessing campus sustainability worldwide.
- To inform governments, environmental agencies, and the general public about the sustainability programs adopted by each campus.

The UI classification makes it possible to account for widely differing understandings of sustainability. The universities analyzed revealed major differences not only in terms of their awareness and commitment to this cause, but also in terms of the budget they can allocate to it. Hence, the ranking is based on a Total Score with a maximum value of 10,000 points, representing the sum of six indicators weighted according to their relevance in the final calculation.

- Setting & Infrastructure: provides information on the environmental policy adopted by the institution to foster active involvement in the protection of the environment and the development of sustainable energies. Assigned a global weighting of 15% and defined by:

    - Outdoor Surface/Total Surface (3%)
    - Outdoor Surface/Campus Population (3%)
    - Campus area covered with forest vegetation (2%)
    - Campus area covered with cultivated vegetation (2%)
    - Campus surface with water-absorbing capacity (3%)
    - University budget allocated to sustainability (2%)

- Energy & Climate Change: explores the application of renewable and efficient energy in university buildings as well as the level of knowledge about nature and energy resources. This is considered the most relevant indicator in the index. It is assigned a global weighting of 21% and defined by:

    - Use of energy-efficient appliances (2%)
    - Implementation of intelligent buildings (3%)
    - On-campus renewable energy production (3%)
    - Total Energy Consumption/Campus Population (3%)

- Renewable energy production/energy consumption (2%)
- Green Building Implementation Element (3%)
- Program for the reduction of greenhouse gas emissions (2%)
- Total Carbon Footprint/Campus Population (3%)

- Waste: evaluates the waste treatment programs that have been implemented on campus. Assigned a global weighting of 18% and defined by:

  - Program to reduce the consumption of paper and plastic on campus (3%)
  - University Waste Recycling Program (3%)
  - Toxic waste management (3%)
  - Treatment of organic waste (3%)
  - Inorganic waste treatment (3%)
  - Wastewater disposal (3%)

- Water: assesses the water consumption as well as water environment conservation and protection programs. Assigned a global weighting of 10% and defined by:

  - Water Conservation Program (3%)
  - Water Recycling Program (3%)
  - Use of water-efficient appliances (2%)
  - Consumption of piped water (2%)

- Transportation: evaluates the transportation policies aimed at limiting the number of vehicles on campus as well as promoting the use of public transport or cycling as better alternatives. All of this plays an important role in reducing carbon emissions, and therefore, the level of pollution at the university. Assigned a global weighting of 18% and defined by:

  - Vehicles/Campus Population (2%)
  - Transfer Services/Campus Population (2%)
  - Bicycles/Campus Population (2%)
  - Types of parking areas (2%)
  - Transportation initiatives to reduce the number of private vehicles on campus (2%)
  - Reduction of parking areas for private vehicles in the last 3 years (2%)
  - Relocation services (3%)
  - Pedestrian and bicycle policy on campus (3%)

- Education & Research: assesses the role of the university as a learning center for society on sustainability issues. Assigned a global weighting of 18% and defined by:

  - Subjects on sustainability/Total subjects (3%)
  - Investment in sustainability research/Total investment in research (3%)
  - Sustainability publications (3%)
  - Sustainability events (3%)
  - Student organizations related to sustainability (3%)
  - Sustainability websites (3%)

Based on these six categories, we performed a cluster analysis for the universities that participated in the 2018 UI GreenMetric. As indicated in the methodology section, Ward's Method was applied to create four distinct groups of higher education institutions, defined according to their level of involvement in the different aspects of sustainability: high, medium-high, medium-low, and low.

Figure 1 shows the distribution of the 719 universities according to the degree of sustainability achieved. It shows that the distribution was not balanced in terms of the number of universities that made up each group.

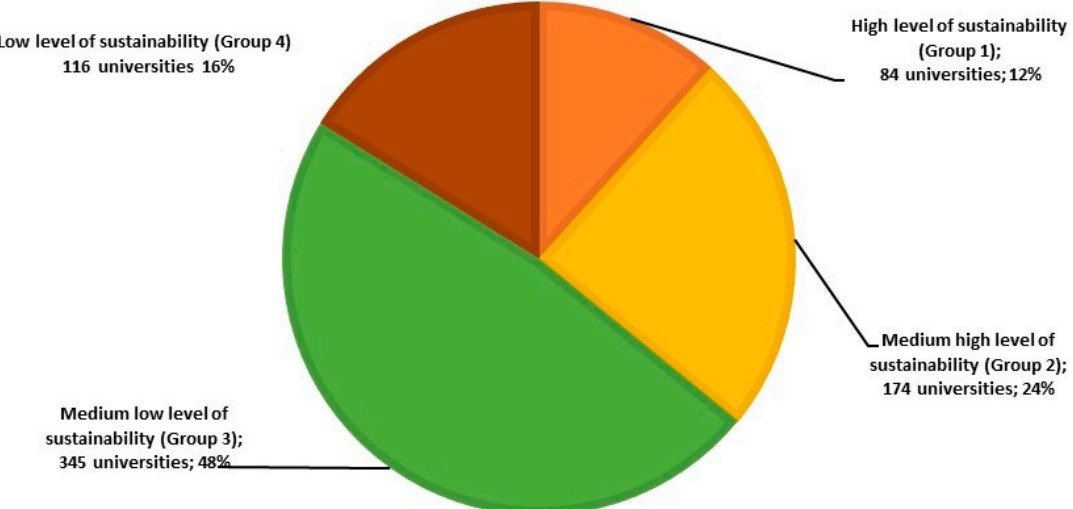

**Figure 1.** Distribution of universities by cluster.

According to the results of the cluster analysis, the largest cluster corresponded to universities with a medium-low level of sustainability (48% of all universities analyzed). This result indicates that many higher education institutions still have to make substantial improvements in this area. In second place was the medium-high group, representing 24% of the total, followed by the least sustainable (16%), and, last, the group of campuses that showed the greatest commitment to addressing all aspects of sustainability, comprising 12% of all of the universities analyzed. Leaders in higher education institutions are striving to turn these results around and to achieve greater involvement and commitment to the sustainability policies currently being implemented around the world. Table 1 presents the main statistics that allow us to characterize the features of each group.

Group 1, the high-sustainability group, was composed of the universities that had achieved the highest scores in all of the aspects covered by the UI GreenMetric index. The Total Scores in this group ranged between 9125 points, registered by Wageningen University & Research (the Netherlands), and 6350 points reached by the National Cheng Kung University (China). The 84 institutions that made up this group were spread across 21 countries; the largest number was found in the United States (10), followed by the United Kingdom (9) and China-Taipei (7). In addition, there were six campuses that achieved the maximum score in the Waste category (1800 points), seven in Water (1000 points), and three in Education & Research (1800 points). Conversely, the lowest scores on average were registered for the aspects related to water treatment and infrastructure. Some universities did not even reach half the maximum possible score for these two attributes.

Groups 2 and 3, which represented the intermediate sustainability levels, were the largest, and contained 24% and 48% of the universities analyzed, respectively. In the medium-high group, some campuses had achieved the maximum score in the same aspects as universities in Group 1, however, this group showed greater neglect of the issues related to energy, transportation, and infrastructure. These universities came from 47 countries, with the United States once again predominating (21), followed by Spain (19), and Colombia (11). For its part, the medium-low group displayed a greater neglect of the issues related to the treatment of water and waste. This group contained universities from 62 different countries, most notably Indonesia (32), the United States (29), and Russia (28).

**Table 1.** Main statistics by homogeneous groups of universities.

| | Group 1: High Level of Sustainability (84 Universities) | | | | | |
|---|---|---|---|---|---|---|
| | **Energy & Climate Change** | **Waste** | **Water** | **Setting & Infrastructure** | **Transportation** | **Education & Research** |
| **Mean** | 1345.5 | 1450.8 | 729.7 | 1121.1 | 1363.6 | 1460.4 |
| **Max** | 1800.0 | **1800.0** | **1000.0** | 1450.0 | 1700.0 | **1800.0** |
| **Min** | 900.0 | 900.0 | 300.0 | 625.0 | 1000.0 | 1050.0 |
| **St. Error** | 215.0 | 208.4 | 158.5 | 170.8 | 145.1 | 168.3 |
| | Group 2: Medium-High Level of Sustainability (174 Universities) | | | | | |
| | **Energy & Climate Change** | **Waste** | **Water** | **Setting & Infrastructure** | **Transportation** | **Education & Research** |
| **Mean** | 1087.3 | 1169.8 | 559.6 | 859.4 | 1001.5 | 1156.9 |
| **Max** | 1700.0 | **1800.0** | **1000.0** | 1325.0 | 1600.0 | **1800.0** |
| **Min** | 550.0 | 525.0 | 100.0 | 225.0 | 550.0 | 625.0 |
| **St. Error** | 235.7 | 266.4 | 163.6 | 208.3 | 194.3 | 230.2 |
| | Group 3: Medium-Low Level of Sustainability (345 Universities) | | | | | |
| | **Energy & Climate Change** | **Waste** | **Water** | **Setting & Infrastructure** | **Transportation** | **Education & Research** |
| **Mean** | 831.7 | 775.0 | 355.3 | 798.7 | 768.1 | 853.5 |
| **Max** | 1400.0 | 1575.0 | 850.0 | 1400.0 | 1300.0 | 1475.0 |
| **Min** | 150.0 | 0.0 | 0.0 | 200.0 | 200.0 | 75.0 |
| **St. Error** | 230.6 | 276.8 | 155.6 | 261.9 | 203.6 | 225.3 |
| | Group 4: Low Level of Sustainability (116 Universities) | | | | | |
| | **Energy & Climate Change** | **Waste** | **Water** | **Setting & Infrastructure** | **Transportation** | **Education & Research** |
| **Mean** | 592.6 | 318.7 | 171.1 | 496.5 | 482.1 | 508.8 |
| **Max** | 1325.0 | 750.0 | 675.0 | 1125.0 | 1125.0 | 1050.0 |
| **Min** | 50.0 | 0.0 | 0.0 | 0.0 | 0.0 | 0.0 |
| **St Error** | 278.7 | 190.4 | 145.3 | 225.2 | 214.2 | 233.1 |

Finally, the low sustainability group, Group 4, consisted of 116 universities. Their Total Scores assigned by the UI GreenMetric ranged between 3725 points for the University of Diyala (Iraq) and 1025 points for the University of Applied Science and Technology (Iran). In this group, there were 26 universities that scored zero in the Water category, 15 failed to score in Waste, two in Education & Research, and one in Setting & Infrastructure as well as in Transportation. Overall, this group was weak in all aspects of sustainability. Together, Indonesia (20), Pakistan (20), and Russia (10) accounted for almost half of the universities in this group.

In short, the mean value for each UI GreenMetric category corresponding to the groups formed by the cluster analysis confirmed that the groups displayed sufficient within-group homogeneity for the correct application of DEA (Figure 2).

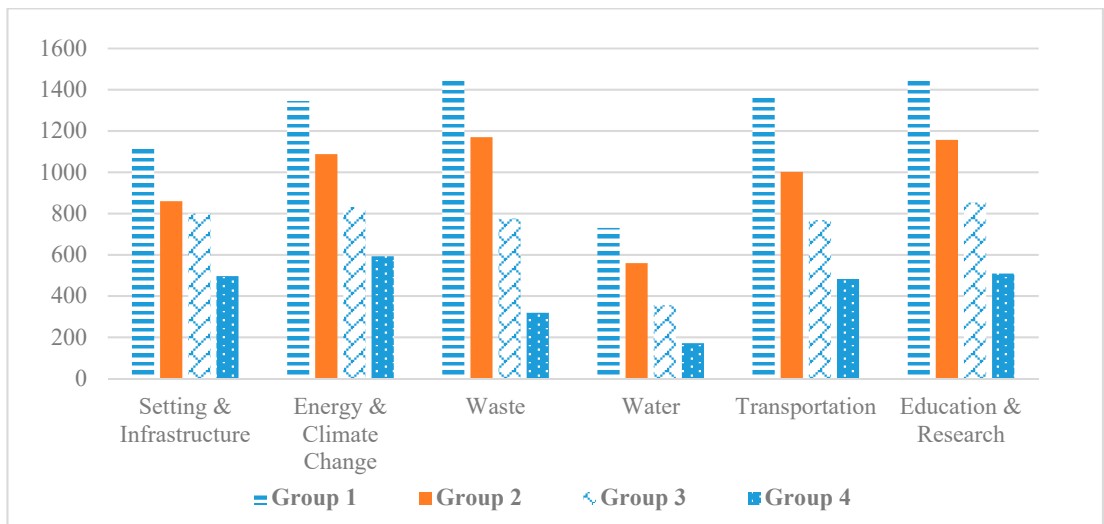

**Figure 2.** Mean value of each UI GreenMetric category by group.

## 4. Results of the Synthetic DEA-GreenMetric Indicator

The synthetic indicator DEA-GreenMetric is constructed from the six UI GreenMetric categories described in the previous section. The aim of the indicator is to produce a ranking of the universities based on their degree of commitment to all of the aspects related to sustainability. The non-parametric DEA technique applied used the following categories as inputs: Setting & Infrastructure, Transportation, and Education & Research. The sum of their weights represented 51% of the UI GreenMetric Total Score, while the remaining 49% was accounted for by the variables taken as outputs (Energy & Climate Change, Waste, and Water).

In the field covered by this research, deciding which variables are input and which are output involves a somewhat arbitrary choice as we were not dealing with a production function. Thus, after carrying out several tests and verifying that the results did not change significantly, we selected the attributes most closely related to the environment for use as outputs. On the other hand, in order to construct the index, we had to transform the variables used as inputs into negative variables or "aspects to improve". This was done by taking the difference between the maximum possible score for each one and the actual score obtained. Thus, the underlying idea is that more tangible elements give rise to certain sustainable practices or activities such as greener spaces, the use of public transport, or courses on the subject of sustainability, which help to achieve more limited use of electricity, reduce the consumption of paper and plastics, and ensure that organic waste is dealt with appropriately.

The synthetic indicator was calculated under variable returns to scale for each of the four groups of universities identified by the cluster analysis (High, Medium-High, Medium-Low, and Low sustainability). Table 2 shows the number of efficient universities in each group (last row) and their home country (first column); as such, it is possible to determine the number of efficient higher education centers in a country as a share of the total number of participating universities in that country (column 7). The results show that in Group 1, 20 universities obtained a score of 1 in their synthetic DEA-GreenMetric indicator, representing 23.80% of their group. In Groups 2, 3, and 4, 24, 35, and 28 campuses achieved a score of 1, representing 13.79%, 10.14%, and 24.13% of the total of their groups, respectively. In short, it can be concluded that the groups classified as high or low sustainability contained a higher proportion of fully efficient campuses. In other words, with their given input use, they managed to maximize the variables of Energy & Climate Change, Waste, and Water within their level of sustainability.

**Table 2.** Distribution of the number of efficient universities by country and group.

| Country | Group 1 High | Group 2 Medium-High | Group 3 Medium-Low | Group 4 Low | Total efficient universities | Efficient Univ./Participating Univ. * |
|---|---|---|---|---|---|---|
| US | 3 | 3 | 5 | 0 | 11 | 18.0% |
| UK | 6 | 2 | 0 | 0 | 8 | 61.5% |
| Thailand | 0 | 0 | 4 | 2 | 6 | 18.7% |
| Indonesia | 1 | 0 | 1 | 4 | 6 | 9.0% |
| Colombia | 0 | 1 | 4 | 0 | 5 | 13.5% |
| Pakistan | 0 | 0 | 1 | 4 | 5 | 14.7% |
| Russia | 0 | 0 | 4 | 1 | 5 | 11.9% |
| Spain | 1 | 3 | 1 | 0 | 5 | 17.8% |
| Mexico | 0 | 2 | 0 | 2 | 4 | 30.7% |
| Turkey | 0 | 1 | 2 | 1 | 4 | 13.3% |
| Germany | 3 | 0 | 0 | 0 | 3 | 30.0% |
| Italy | 0 | 1 | 2 | 0 | 3 | 11.5% |
| Malaysia | 0 | 1 | 2 | 0 | 3 | 16.6% |
| Poland | 0 | 0 | 0 | 3 | 3 | 42.8% |
| Romania | 0 | 0 | 2 | 1 | 3 | 37.5% |
| Brazil | 1 | 0 | 0 | 1 | 2 | 8.7% |
| China | 1 | 0 | 1 | 0 | 2 | 100.0% |
| China Taip | 0 | 0 | 2 | 0 | 2 | 6.9% |
| Ireland | 2 | 0 | 0 | 0 | 2 | 50.0% |
| India | 0 | 0 | 1 | 1 | 2 | 7.6% |
| Japan | 0 | 2 | 0 | 0 | 2 | 20.0% |
| Netherlands | 2 | 0 | 0 | 0 | 2 | 40.0% |
| Saudi Arab. | 0 | 2 | 0 | 0 | 2 | 66.6% |
| Argentina | 0 | 0 | 0 | 1 | 1 | 33.3% |
| Czech Rep. | 0 | 1 | 0 | 0 | 1 | 33.33% |
| Costa Rica | 0 | 0 | 1 | 0 | 1 | 100.0% |
| Denmark | 0 | 1 | 0 | 0 | 1 | 100.0% |
| Ecuador | 0 | 0 | 1 | 0 | 1 | 33.3% |
| Finland | 0 | 1 | 0 | 0 | 1 | 25.0% |
| France | 0 | 1 | 0 | 0 | 1 | 14.2% |
| Iraq | 0 | 0 | 0 | 1 | 1 | 20.0% |
| Hungary | 0 | 0 | 0 | 1 | 1 | 11.1% |
| Jordan | 0 | 1 | 0 | 0 | 1 | 11.1% |
| Kazakhstan | 0 | 0 | 1 | 0 | 1 | 9.0% |
| Philippines | 0 | 0 | 0 | 1 | 1 | 16.7% |
| Portugal | 0 | 1 | 0 | 0 | 1 | 25.0% |

**Table 2.** *Cont.*

| Country | Group 1 High | Group 2 Medium-High | Group 3 Medium-Low | Group 4 Low | Total efficient universities | Efficient Univ./Participating Univ. * |
|---|---|---|---|---|---|---|
| **Slovenia** | 0 | 0 | 0 | 1 | 1 | 33.3% |
| **Sweden** | 0 | 0 | 0 | 1 | 1 | 25.0% |
| **Syria** | 0 | 0 | 0 | 1 | 1 | 100.0% |
| **Tunisia** | 0 | 0 | 0 | 1 | 1 | 50.0% |
| **TOTAL** | 20 (23.8%) | 24 (13.7%) | 35 (10.1%) | 28 (24.1%) | 107 (14.8%) | |

Note (*): Efficient univ/Participating univ = Efficient universities of the country j/total number of participating universities in country j.

In the area of sustainable practices, US and British universities topped the ranking, with a higher number of efficient universities than the remaining 38 countries. However, there were significant differences between them: while the United States registered the highest number of efficient institutions in absolute terms (11 universities), British universities outnumbered their American counterparts in the high sustainability group.

Looking at the distribution by country, it can also be seen that all universities in China, Denmark, Costa Rica, and Syria were assessed as fully efficient within their group. Furthermore, it should be noted that these countries are home to only a few participating campuses, they all implement good sustainability management relative to the level assigned to them. Therefore, they should consider setting policies that will enable them to make progress in this area and thus achieve a higher-level of classification.

Table 3 shows the top five universities according to DEA-GreenMetric and their corresponding UI GreenMetric rankings. The first column shows the ranking of each group according to the Total Score assigned by UI and the second column shows the ranking according to the criterion of the proposed index. The CE results, which enabled the ranking of all of the efficient campuses, are shown in column 4. Lastly, column 5 shows the number of times an efficient university has served as a reference for the inefficient universities in the same group. The results confirmed that in almost all groups, the top three universities have most often served as a reference for the others. Specifically, Universita degli Studi Dell'aquila in Group 3 was the university of reference to 175 universities, that is to say, more than half of the campuses in its group, Wageningen University & Research (Group 1) to 80.9%, King Abdulaziz University (Group 2) to 70.1%, and the University of Central Punjab (Group 4) to 36.2%.

**Table 3.** The UI GreenMetric versus the DEA-GreenMetric for the top five universities.

| Group 1 | | | | |
|---|---|---|---|---|
| Ranking UI GreenMetric | Ranking DEA-GreenMetric | 5 Best Universities | CE | N° Reference |
| 1 | 1 | Wageningen University & Research | 0.991 | [68] |
| 3 | 2 | University of California Davis | 0.939 | [10] |
| 10 | 3 | University of North Carolina Chapel Hill | 0.879 | [3] |
| 6 | 4 | Umwelt-Campus Birkenfeld | 0.867 | [5] |
| 4 | 5 | University of Oxford | 0.862 | [1] |

**Table 3.** *Cont.*

| Group 2 | | | | |
|---|---|---|---|---|
| **Ranking UI GreenMetric** | **Ranking DEA-GreenMetric** | **5 Best universities** | **CE** | **N° reference** |
| 1 | 1 | King Abdulaziz University | 0.967 | [122] |
| 6 | 2 | Inseec U | 0.897 | [67] |
| 2 | 3 | Czech University of Life Sciences Prague | 0.887 | [35] |
| 9 | 4 | Universidad Autónoma de Nuevo León | 0.871 | [7] |
| 8 | 5 | University of Lincoln | 0.868 | [1] |
| **Group 3** | | | | |
| **Ranking UI GreenMetric** | **Ranking DEA-GreenMetric** | **5 Best universities** | **CE** | **N° reference** |
| 1 | 1 | Universita degli Studi dell'Aquila | 0.919 | [175] |
| 3 | 2 | Webster University | 0.895 | [171] |
| 12 | 3 | Universitat de Vic | 0.889 | [163] |
| 4 | 4 | Al-Farabi Kazakh National University | 0.851 | [43] |
| 26 | 5 | Far Eastern Federal University | 0.832 | [76] |
| **Group 4** | | | | |
| **Ranking UI GreenMetric** | **Ranking DEA-GreenMetric** | **5 Best Universities** | **CE** | **N° reference** |
| 3 | 1 | Universitas Maritim Raja Ali Haji | 0.866 | [38] |
| 2 | 2 | University of Central Punjab | 0.861 | [42] |
| 1 | 3 | University of Diyala | 0.848 | [18] |
| 6 | 4 | University of Trnava | 0.827 | [25] |
| 4 | 5 | Agricultural University of Cracow | 0.800 | [19] |

Note: CE, cross-efficiency.

In addition, the results indicate that the scores assigned by the two indices were very similar. The top five universities in each group according to the proposed indicator (DEA-GreenMetric) also held high-ranking positions in the UI GreenMetric, except in Group 3, where there were some discrepancies. The coefficient of correlation between the rankings of the two indices corroborates the close relationship between them (Table 4).

**Table 4.** Coefficient of correlation between the UI GreenMetric and DEA-GreenMetric by group for efficient observations.

| | **Coefficient Correlation** |
|---|---|
| **Group 1** | 0.730 |
| **Group 2** | 0.771 |
| **Group 3** | 0.669 |
| **Group 4** | 0.788 |

The DEA-GreenMetric indicator provides a more complete classification of the participating campuses, as the UI GreenMetric assigns some institutions the same Total Score, making it impossible to rank them all. This is due to the weighting system established for the construction of the index, where individual weights are assigned to each of the analyzed variables, which are then summed to calculated the global value. Table 5 identifies the efficient universities allocated the same Total Score by the UI GreenMetric (columns 3 and 4). It also shows how the proposed index assigns an objective ranking (columns 5 and 6).

**Table 5.** Ranking of the universities with the same Total Score.

| Group 1 | Country | UI GreenMetric Total Score | Ranking UI GreenMetric | CE | Ranking DEA-GreenMetric |
|---|---|---|---|---|---|
| Shandong Normal University—Lishan College | China | 7975 | 13 | 0.815 | 11 |
| Universidad de Alcala | Spain | 7975 | 13 | 0.738 | 18 |
| Dublin City University | Ireland | 8025 | 11 | 0.868 | 6 |
| Keele University | UK | 8025 | 11 | 0.814 | 12 |
| Umwelt-Campus Birkenfeld | Germany | 8350 | 6 | 0.867 | 7 |
| University of Groningen | Netherland | 8350 | 6 | 0.860 | 9 |
| **Group 2** | **Country** | **UI GreenMetric Total Score** | **Ranking UI GreenMetric** | **CE** | **Ranking DEA-GreenMetric** |
| Aalborg University | Denmark | 7050 | 7 | 0.858 | 7 |
| Universidad de Bogotá Jorge Tadeo Lozano | Colombia | 7050 | 7 | 0.849 | 9 |
| Czech University of Life Sciences Prague | Czech Rep | 7275 | 2 | 0.887 | 3 |
| Shinshu University | Japan | 7275 | 2 | 0.787 | 13 |
| **Group 3** | **Country** | **UI GreenMetric Total Score** | **Ranking UI GreenMetric** | **CE** | **Ranking DEA-GreenMetric** |
| Jabatan Pendidikan Politeknik Malaysia | Malaysia | 4825 | 35 | 0.629 | 26 |
| Universidad San Francisco de Quito | Ecuador | 4825 | 35 | 0.628 | 27 |
| Universidad Católica de Oriente | Colombia | 4925 | 31 | 0.583 | 30 |
| Universita della Calabria | Italy | 4925 | 31 | 0.428 | 34 |
| Far Eastern Federal University | Russia | 5050 | 26 | 0.832 | 5 |
| Minin University | Russia | 5050 | 26 | 0.721 | 18 |
| Mehran University of Engineering & Technology | Pakistan | 5050 | 26 | 0.612 | 29 |
| University of Phayao | Thailand | 5075 | 25 | 0.656 | 24 |
| North Eastern University | Thailand | 5075 | 25 | 0.629 | 25 |
| Ege University | Turkey | 5075 | 25 | 0.563 | 32 |
| Universitat de Vic—Universitat Central de Catalunya | Spain | 5425 | 12 | 0.889 | 3 |
| Shinawatra University | Thailand | 5425 | 12 | 0.732 | 17 |
| Universidad Tecnologica de Pereira | Colombia | 5550 | 8 | 0.780 | 10 |
| East Stroudsburg University | US | 5550 | 8 | 0.777 | 11 |
| **Group 4** | **Country** | **UI GreenMetric Total Score** | **Ranking UI GreenMetric** | **CE** | **Ranking DEA-GreenMetric** |
| Polish Japanese Institute of Information Technology in Warsaw | Poland | 2875 | 23 | 0.617 | 16 |
| Universitas Muhammadiyah Surakarta | Indonesia | 2875 | 23 | 0.591 | 17 |
| Swedish Defence University | Sweden | 3000 | 18 | 0.776 | 8 |
| University at Bialystok | Poland | 3000 | 18 | 0.715 | 13 |
| Universidade Federal Do Abc Ufabc | Brazil | 3350 | 8 | 0.735 | 11 |
| Jawaharlal Institute of Postgraduate Medical Education & Research | India | 3350 | 8 | 0.661 | 14 |
| University of Sindh Jamshoro | Pakistan | 3350 | 8 | 0.556 | 19 |
| University of Trnava | Slovenia | 3400 | 6 | 0.828 | 4 |
| Silpakorn University | Thailand | 3400 | 6 | 0.752 | 10 |

Note: CE = cross-efficiency.

In line with the initial research objectives, having determined the DEA results for each university, we attempted to identify whether the mean inputs and outputs of the observations closest to the frontier were statistically different from those farthest away. To do so, we used the Kruskal–Wallis test. A synthetic indicator value of 0.8 was taken as the cut-off value to divide the groups, attempting to identify the value that best fit the results obtained (a threshold of 0.8 was set in order to have samples of approximately the same size in terms of the number of observations) (Table 6).

**Table 6.** Results of the Kruskal–Wallis Test.

|  |  | Energy & Climate Change | Waste | Water | Setting & Infrastructure | Transportation | Education & Research |
|---|---|---|---|---|---|---|---|
| **GROUP 1** | Mean (Ef < 0.8) | 1131.9 | 1200.0 | 1241.7 | 625.0 | 1375.0 | 1437.5 |
|  | Mean (Ef > 0.8) | 1118.2 | 1385.2 | 1508.0 | 758.3 | 1360.6 | 1466.7 |
|  | Chi-Square | 9.960 | 19.928 | 11.420 | 0.285 | 0.210 | 0.280 |
|  | *p*-value | 0.001 | 0.000 | 0.000 | 0.593 | 0.647 | 0.597 |
| **GROUP 2** | Mean (Ef < 0.8) | 848.4 | 977.5 | 1065.0 | 477.5 | 955.6 | 1136.6 |
|  | Mean (Ef> 0.8) | 868.9 | 1180.9 | 1259.0 | 629.5 | 1040.7 | 1174.2 |
|  | Chi- Square | 30.788 | 19.597 | 42.347 | 0.538 | 7.053 | 0.720 |
|  | *p*-value | 0.000 | 0.000 | 0.000 | 0.463 | 0.007 | 0.396 |
| **GROUP 3** | Mean (Ef < 0.8) | 769.0 | 750.1 | 690.0 | 315.8 | 730.0 | 834.2 |
|  | Mean (Ef > 0.8) | 844.8 | 958.7 | 907.2 | 416.9 | 827.4 | 883.7 |
|  | Chi- Square | 65.112 | 43.332 | 31.508 | 6.029 | 17.637 | 1.812 |
|  | *p*-value | 0.000 | 0.000 | 0.000 | 0.014 | 0.000 | 0.178 |
| **GROUP 4** | Mean (Ef < 0.8) | 483.2 | 481.6 | 256.6 | 112.9 | 428.5 | 469.5 |
|  | Mean (Ef > 0.8) | 513.0 | 729.3 | 395.2 | 242.8 | 548.1 | 557.2 |
|  | Chi- Square | 22.214 | 13.896 | 20.245 | 0.302 | 7.674 | 3.563 |
|  | *p*-value | 0.000 | 0.000 | 0.000 | 0.582 | 0.005 | 0.059 |

As can be seen from Table 6, the results for the three outputs were similar in all groups (significant chi-square, *p*-value < 0.05); that is, regarding the environmental aspects (Energy & Climate Change, Waste, and Water), there were significant differences within each group according to whether the efficiency level was above or below 0.8. Conversely, within-group differences were negligible for all of the inputs except Transportation, which was significant (that is, it differed between the established subsamples) in all groups apart from the high-sustainability universities.

Finally, we used the value of the universities' targets to determine the degree of change required in the campus sustainability categories in order for them to improve their position in the ranking (Table 7).

**Table 7.** Potential Improvement of Inputs/Ouputs.

|  | Ouputs | | | Inputs | | |
|---|---|---|---|---|---|---|
|  | Energy & Climate Change | Waste | Water | Setting & Infrastructure | Transportation | Education & Research |
| **GROUP 1** | 30.9% | 24.0% | 34.3% | 40.6% | 38.8% | 88.2% |
| **GROUP 2** | 39.4% | 36.1% | 34.3% | 12.0% | 13.1% | 23.0% |
| **GROUP 3** | 41.6% | 54.0% | 43.8% | 22.3% | 13.6% | 11.4% |
| **GROUP 4** | 68.7% | 68.4% | 74.7% | 10.0% | 4.4% | 9.6% |

Again, the results in Table 7 indicate that campuses should make greater efforts to improve their performance regarding environmental variables. The most notable situation is in Group 4 (low sustainability), where an increase of almost 69% is required for the items relating to energy and recycling, and more than 74% for those related to water treatment and use, in order for them to be among the best in their cluster. Conversely, the universities classified as high sustainability should enact policies that would promote education in that area (88.24%), without overlooking other aspects such as infrastructure or transport-related items (40.67% and 38.84%, respectively).

## 5. Discussion and Conclusions

This article proposed a synthetic indicator constructed from the variables that compose the UI GreenMetric. The aim was to produce a ranking of the 719 universities that participated in the 2018 UI GreenMetric, and thereby obtain information to complement that provided by UI. First, clustering

methodology was used to ensure the homogeneity of the samples and thus provide optimal conditions for the application of DEA.

The cluster analysis allowed us to identify four levels of sustainability representing the different degrees of commitment shown by the campuses: high, medium-high, medium-low, and low sustainability, with the latter two groups representing 64% of the total sample. The statistics for these groups revealed that the institutions that achieved the lowest scores in sustainability should take stronger measures in all of the variables analyzed, but particularly to the treatment of water and waste. On the other hand, the most committed universities (high and medium-high) managed to obtain maximum scores in the treatment of water and waste as well as in research and other related educational aspects. These conclusions were also corroborated by the Kruskal–Wallis test, the results of which revealed that the environmental variables (Water, Waste, and Energy & Climate Change) represented the categories that set the highest-ranking institutions in each group apart from the rest.

The DEA-GreenMetric contributes to the literature by enabling a ranking of universities according to the intensity of their efforts to manage environmental concerns and sustainable development. The application of the DEA technique assigns maximum scores to a number of universities, ranking the rest according to their relative efficiency levels. The possible ambiguity resulting from the classification of universities located on the efficiency frontier (efficiency scores of 1) was resolved by applying CE, which enabled a complete ranking of all universities.

In short, this research has yielded a synthetic indicator to complement the information provided by the UI GreenMetric; the methodology used to produce the latter means that two institutions can obtain the same score. The results for the top five universities in each group revealed a degree of similarity between the UI GreenMetric and the indicator created. Furthermore, it can be concluded that sustainability efforts focused on environmental variables are the least well-established in these institutions. Universities should thus take measures to improve their performance in all issues related to waste management and water as well as energy and climate change.

Subsequent publications of the UI GreenMetric report will allow these results to be updated in order to assess the progress made by the universities analyzed. The greater public awareness of sustainability calls for a high degree of transparency so that the policies being implemented can be clearly understood and serve as a model for institutions that are lagging behind.

**Author Contributions:** All authors contributed equally to this work. They had read and approved the manuscript.

**Funding:** This research received no external funding

**Conflicts of Interest:** The authors declare no conflicts of interest

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
