# Peer review of "Sustainability in Universities: DEA-GreenMetric"

_sustainability, doi:10.3390/su11143766_

Round 1

Reviewer 1 Report

I think the paper is very well made it. 

The introduction creat a good image of previous research about this topic.

The methodology  is well explained. In methodology was introduced explanations for the concept which sustain this new indicators proposed by this research.

There is statistics analyses. 

Conclussions are relevant. 

Author Response

We are very grateful for all the favourable comments made by the reviewer.

Reviewer 2 Report

The subject of measuring the sustainability of universities to identify areas that are working well and others requiring improvement, is both important and interesting. 

However I have some hesitation in accepting the paper as it stands for several reasons: the potted history of sustainability is unconvincing; the importance of considering sustainability in comparing universities is not made with great depth of understanding or conviction; the use of DEA analysis may be appropriate but is not clearly explained to this reader - who is neither an economist nor statistician though reasonably well-versed in sustainability; I do not understand the problem of UI Green Metric allowing two institutions to achieve the same score (line 392); seemingly random use of accuracy of the numbers expressed in the analysis tables from none to one to two and sometimes three decimal points, and perhaps carelessly indicated by either a full stop or a comma (see Tables 1, 2, 7); repeated first line of data in Table 1 group 1; Table 2 needs clarification of the data shown, especially the denominator used for the last column. Further some details of English (eg 'indices' not 'indexes', line 48) and general expression, need editing for example to be consistent in the use of tenses which varies between past, present and future tense in a manner that is not consistent; use of an alternate term for 'own elaboration' as the source of data in tables and figures. It would be helpful to mention the number of universities globally as part of the description of UI Green Metric (line 182, or line 22).

Some aspects of university sustainability that appear in the broader discussion on sustainability of universities, covers attributes that are not mentioned in this paper. For example, clarifying if the discussion covers embedded energy in capital investment in buildings and equipment as well as operational energy (around line 200); whether to differentiate between renewable vs non-renewable energy sources; whether to use a single metric such as CO2/person; whether  to include CO2 emissions from staff and student travel to and from the university, and to and from the sites of their research or conferences, is commonly mentioned. Indeed that paper mentions that 'the choice of variables is somewhat arbitrary' (line 282).

Author Response

We are very grateful for all the comments made by the reviewer. All the reviewer's comments have been answered in an attached file.

Reviewer 3 Report

The article reviews the UI Green Metric classification by applying DEA statistical methodology, obtaining results of great interest that question part of the results published by Universitas Indonesia. The article does not present a new methodology; however, it provides a critical view of the evaluation system, resulting of great interest for its publication. I even consider that authors should communicate their results to the institution promoting the evaluation and share with them the rectified evaluation system.

In my opinion, as a point that could strengthen the work done is that; although the analysis has been done with the 2018 results and is expected to be reviewed annually, the methodology has not incorporated the evolution of the data from previous years. As a comment to the authors, it would be very interesting to include in the statistical assessment the data published for those universities that were integrated into the evaluation system before 2018.

Considering the article acceptable for publication, I would like to add some comments:

Perhaps a deeper review of campus sustainability should be done. I would recommend that the introduction section include references that cover different concepts and methodologies for analyzing sustainability in higher education. Sustainability in higher education has had many approaches for years. It is not a recent topic and it draws new knowledge since the Agendas 21 were developed. Sustainability is part of the institutional vision and practice in universities even before the 21st century. I refer to authors such as:

-Van Weenen, H., 2000. Towards a vision of a sustainable university. International Journal of Sustainability in Higher Education, 1(1), pp. 20-34

-Sharp, L., 2002. Green campuses: The road from little victories to systemic transformation. International Journal of Sustainability in Higher Education, 3(2), pp. 128-145

-Shriberg, M., 2002. Institutional assessment tools for sustainability in higher education: Strengths, weaknesses, and implications for practice and theory. International Journal of Sustainability in Higher Education, 3(3), pp. 254-270

I also miss literature related to very complex aspects of evaluation and that affect the assessment indexes, for example about transport issues:

-Balsas, C.J.L., 2003. Sustainable transportation planning on college campuses. Transport Policy, 10(1), pp. 35-49.

On the participation of stakeholders in the development of sustainability:

-Ribalaygua Batalla, C., García Sánchez, F., 2016. Creating a Sustainable Learning District by Integrating Different Stakeholders’ Needs. Methodology and Results from the University of Cantabria Campus Master Plan. In: Leal Filho W., Brandli L. (eds) Engaging Stakeholders in Education for Sustainable Development at University Level. World Sustainability Series. Springer, Cham.

- Dagiliūtė, R., Liobikienė, G., Minelgaitė, A., 2018. Sustainability at universities: Students’ perceptions from Green and Non-Green universities. Journal of Cleaner Production, 181, pp. 473-482.

Or the analysis on the relations in open spaces as:

 - Chapman, M.P., 2006. American places: in search of the twenty-first century campus. Praeger Publishers, Westport.

Also documents with a holistic vision related to sustainability on campus:

-Alshuwaikhat, H.M., Abubakar, I., 2008. An integrated approach to achieving campus sustainability: assessment of the current campus environmental management practices. Journal of Cleaner Production, 16(16), pp. 1777-1785

Related to the DEA methodology, and about the information related to Table 3 I miss references of interest such as:

- Adler, N., Friedman, L., Sinuany-Stern, Z., 2002. Review of ranking methods in the data envelopment analysis context. European Journal of Operational Research 140, 249–265

The authors should provide the resulting values related to CE calculation methodology in a table or in an attached file. It is difficult to follow the discourse without CE data.

In order to improve the quality of the article, allow me to add some recommendations and communicate some deficiencies detected:

- Review British or American style (-ized) (ised)

- In line 187 reference 49 does not exist. This is perhaps 48.

- Figure 1 could be remove because all data are explained above.

- In table 1 the firs line “Mean” is repeated.

- In table 6 use “Chi-Square” instead in Spanish

- In Discussion and Conclusion section it would be interesting to define the percentage of overall variation obtained with DEA with respect to the UI Green Metric values.

Author Response

We are very grateful for all the comments made by the reviewer.

In my opinion, as a point that could strengthen the work done is that; although the analysis has been done with the 2018 results and is expected to be reviewed annually, the methodology has not incorporated the evolution of the data from previous years

1.      The results are difficult to compare with previous years because some indicators do not include exactly the same thing, as is the case of "Education & research" where previous years have only taken into account the term education leaving out the content of research. In addition, the sample of universities is not homogeneous from one year to the next, making comparative analysis more difficult.

I would recommend that the introduction section include references that cover different concepts and methodologies for analyzing sustainability in higher education.

2.      The author(s) thank the reviewer for the extensive recommended bibliography. They consider that its incorporation has enriched the bibliographic review of the introduction of the article.

Related to the DEA methodology, and about the information related to Table 3 I miss references of interest.

3.     The reference required by the reviewer has been included.

In order to improve the quality of the article, allow me to add some recommendations and communicate some deficiencies detected:

4.      The reviewer's recommendations have been considered.